

# Overgrazing-induced legacy effects may permit *Leymus chinensis* to cope with herbivory

Fenghui Guo[1,2], Xiliang Li[2], Saheed Olaide Jimoh[2,3], Yong Ding[2], Yong Zhang[2], Shangli Shi[1] and Xiangyang Hou[1,2]

[1] Pratacultural College, Gansu Agricultural University, Lan Zhou, Gan Su Province, China
[2] Institute of Grassland Research, Chinese Academy of Agricultural Sciences, Hohhot, Inner Mongolia, China
[3] Sustainable Environment Food and Agriculture Initiative (SEFAAI), Lagos, Nigeria

## ABSTRACT

There is growing evidence that herbivory-induced legacy effects permit plants to cope with herbivory. However, herbivory-induced defense strategies in plants against grazing mammals have received little attention. To further understand the grazing-induced legacy effects on plants, we conducted a greenhouse experiment with *Leymus chinensis* experiencing different grazing histories. We focused on grazing-induced legacy effects on above-ground spatial avoidance and below-ground biomass allocation. Our results showed that *L. chinensis* collected from the continuous overgrazing plot (OG) exhibited higher performance under simulated grazing in terms of growth, cloning and colonizing ability than those collected from the 35-year no-grazing plot (NG). The enhanced adaptability of OG was attributed to increased above-ground spatial avoidance, which was mediated by larger leaf angle and shorter height (reduced vertical height and increased leaf angle contributed to the above-ground spatial avoidance at a lower herbivory stubble height, while reduced tiller natural height contributed to above-ground spatial avoidance at a higher herbivory stubble height). Contrary to our prediction, OG pre-allocated less biomass to the rhizome, which does not benefit the herbivory tolerance and avoidance of *L. chinensis*; however, this also may reflect a tolerance strategy where reduced allocation to rhizomes is associated with increased production of ramets.

## INTRODUCTION

Environmental disturbances (e.g., drought, herbivory) can have persistent effects on ecological attributes (e.g., ecological processes, community structure, population dynamics, and plant and soil characteristics) long after they occur (i.e., the legacy effect) (*Fox et al., 2015*; *Kafle & Wurst, 2019*). Legacy effects are ubiquitous phenomena in nature and have been extensively studied in the context of plant succession, herbivory, invasive plants, ecosystem engineering, and human land use (*Cuddington, 2011*; *Kostenko et al., 2012*; *Wurst & Ohgushi, 2015*). In grassland ecosystems, herbivory-induced legacy effects on ecological processes, such as plant succession and biological diversity change, can persist for decades

Corresponding author
Xiangyang Hou, houxy@gsau.edu.cn

and even millennia (*Holeski, Jander & Agrawal, 2012*; *Fox et al., 2015*). Some studies have shown that legacies in plant defense strategies can mediate herbivory-induced legacy effects on ecological processes. For instance, defense traits (e.g., chemical defense substances, resource reallocation) can significantly affect plant-herbivore relationships (generally increasing plant adaptations to herbivores), as well as interspecific relationships and soil characteristics (*Wurst & Ohgushi, 2015*). Therefore, clarifying the overgrazing-induced legacy effects in plant defense strategies is critical for understanding processes occurring in grazing ecosystems. However, most previous studies on herbivory-induced legacy effects on plant defense strategies have focused on short-term insect herbivory, in contrast, studies examining long-term livestock grazing have received less attention by comparison (*Holeski, Jander & Agrawal, 2012*; *Kafle & Wurst, 2019*).

Strategies by which plants cope with herbivory include resistance, avoidance, and tolerance (*Didiano et al., 2014*). While resistance strategies (e.g., thorns, higher tannin concentrations) play an important role in coping with insect herbivory (*Agrawal, 2002*; *Holeski, Jander & Agrawal, 2012*), they may be less useful in plants experiencing herbivory from livestock or other grazing mammals, as these large animals are unable to selectively graze at such a fine scale (*Menard et al., 2002*; *Benot et al., 2013a*). Empirical evidence indicates that plants under grazing mammal herbivory show adaptive legacy effects via above-ground spatial avoidance traits which resulted in the distribution of more above-ground biomass close to the ground, including larger leaf angles, shorter height, and more prostrate growth forms (*Polley & Detling, 1988*; *Polley & Detling, 1990*; *Painter, Detling & Steingraeber, 1993*; *Tomás, Carrera & Poverene, 2000*; *Li et al., 2015*; *Ren et al., 2017*). However, studies examining the above-ground biomass vertical distribution have been qualitative—not quantitative. The lack of rigorous quantitative approaches severely limits our understanding of the role of these avoidance traits in the responses of plants to grazing mammal herbivory (e.g., livestock grazing). For example, the specific morphological characters that lead to the near-surface distribution of above-ground biomass to reduce the possibility of defoliation by grazing mammals remain unclear.

Biomass reallocation is a fundamental strategy for plants to cope with herbivory. When subjected to above-ground herbivory, more biomass is mobilized above-ground to facilitate the recovery of growth (*Liu et al., 2018*), and a transient transformation of resources away from herbivores occurs within hours after herbivory (*Anten & Pierik, 2010*; *Orians, Thorn & Gomez, 2011*). Some studies have indicated that plant tolerance is tightly linked to biomass allocation patterns expressed before herbivory; that is, larger belowground biomass pre-allocation is associated with stronger tolerance of plants to above-ground herbivory (*Fornoni, 2011*; *Lurie, Barton & Daehler, 2017*). In addition to the importance of tolerance, large belowground biomass pre-allocation potentially helps plants avoid above-ground herbivory. There is growing evidence that long-term grazing induces higher belowground biomass allocation at the community and population levels (*Milchunas & Lauenroth, 1993*; *Lindwall et al., 2013*). Furthermore, more biomass allocation to roots for plants collected from grazing areas has been observed in common garden environments free of herbivory disturbance (*Jaramillo & Detling, 1988*; *Polley & Detling, 1988*; *Polley & Detling, 1990*). However, few studies have examined herbivory-induced legacy effects on plant

belowground biomass allocation, especially the allocation of belowground reproductive organs such as rhizomes.

*Leymus chinensis*, a rhizomatous clonal plant, is the dominant species on the Inner Mongolia typical steppe grasslands. This grass species is relished by livestock (e.g., cattle, sheep) and has been subjected to overgrazing for more than 50 years. Similar to plants that have been studied in other regions, *L. chinensis* exhibits significant grazing-induced legacy effects, such as short height and short leaves (*Li et al., 2015*; *Ren et al., 2017*). However, no quantitative studies have characterized herbivory-induced legacy effects on the vertical distribution of above-ground biomass and belowground biomass allocation (especially the allocation of rhizome biomass). Although earlier studies (*Oesterheld & McNaughton, 1988*; *Loreti, Oesterheld & Sala, 2001*) have shown that plants subjected to grazing disturbance showed stronger adaptability to herbivory compared with ungrazed plants in a common garden environment, studies reporting legacy effects underlying *L. chinensis* adaptation to grazing are limited.

To further characterize the grazing-induced legacy effects on plants, we conducted a greenhouse pot experiment with *L. chinensis* collected from two adjacent plots separated by a pasture fence. The first was a 35-year no-grazing plot and the second is a long-term overgrazing plot. Our study sought to answer three questions. First, does *L. chinensis* exposed to long-term overgrazing disturbance exhibit enhanced above-ground spatial avoidance (measured by the above-ground biomass vertical distribution)? If so, which individual characteristics contribute to this trait? Second, are there overgrazing-induced legacy effects on *L. chinensis* in terms of the belowground biomass allocation (i.e., root and rhizome)? Third, does the *L. chinensis* collected from the grazing plot exhibit stronger adaptation to simulated herbivory compared with those collected from the no-grazing plot?

## MATERIALS & METHODS

### Materials

#### Description of study sites

Samples of *L. chinensis* were collected from typical steppe grassland located at the Inner Mongolia Grassland Ecosystem Research Station (43°38′N, 116°42′E). The sampling sites comprise two adjacent plots separated by a pasture fence. The first was a no-grazing plot (600×400 m), which has been fenced since 1983 for long-term ecological observations, and the second was a continuously overgrazing plot (600×100 m) that has been grazed at a stocking rate of ∼3 sheep units per hectare for more than 50 years. However, the stocking rate recommended by the local government is ∼1.5 sheep units per hectare to achieve a balance between grassland productivity and livestock forage requirements. Thus, the continuously grazed plot has experienced heavy grazing pressure over the last several decades (*Ma et al., 2019*).

### Materials Collection at the study sites

Genotypes showed different phenotypic plasticity to environmental disturbance and different genetic structures may be one of the main mechanisms mediating overgrazing-induced legacies on *L. chinensis* (*Jonsephs, 2018*). We could not determine the genotype of each *L. chinensis* individual collected from both plots. To reduce the possible impacts of genotypes on our experimental results, we conducted our experiment at the population level and used the high-replication sampling.

We sampled 150-segment rhizomes with a root drill at 150 random points in each of the two treatment plots at the beginning of the growing season. Although we acknowledge pseudoreplication in the experimental design, as each treatment consisted of one large plot with subsamples as replicates, we sampled the entire area for each plot except for the margin and ensured that the distance between every two sampling points was greater than 20 m; this was done both to improve the representativeness of the samples for each plot across each of the large neighboring plots. Each rhizome was approximately four cm long and contained at least one sprouting section containing buds. To prevent the sampled rhizome from losing its vitality following removal from the soil environment, we immediately transferred samples into a container with moistened soil.

### Cultivation in the greenhouse

In the laboratory, each rhizome was dissected into a 2-cm long section with only one node. All of the rhizomes were cultivated in flowerpots, 20 cm in diameter and 15 cm high, which were kept in the greenhouse. Each flowerpot contained 3 kg of soil collected from points adjacent to the experimental site and was subsequently planted with one rhizome section. There were 150 flowerpots for rhizomes collected from the no grazing plot and the overgrazing plot, respectively. After 20 days, the rhizomes had sprouted in approximately 100 flowerpots from each treatment.

## Experimental design & measurements
### Experimental design

The experiment was a full factorial design and consisted of two factors. The first was the source of *L. chinensis* (NG: *L. chinensis* collected from the no-grazing plot; OG: *L. chinensis* collected from the overgrazing plot), and the second was simulated grazing (CK: no simulated grazing; H8: simulated moderate grazing; H4: simulated heavy grazing). For NG and OG, we randomly selected 90 flowerpots that had sprouted rhizomes. These flowerpots were randomly arranged in the greenhouse, and we alternated their position every week to exclude the influence of external factors (e.g., light). One-third of the *L. chinensis* growing in flowerpots (in both NG and OG separately) were randomly assigned to CK, H8, and H4. However, several *L. chinensis* died during the experiment and the numbers left for each treatment were NG*CK (22), NG*H8 (29), NG*H4 (25), OG*CK (22), OG*H8 (26), and OG*H4 (28). We clipped plants with scissors to conduct the simulated grazing as per *Turley et al. (2013)* and *Didiano et al. (2014)*. We simulated moderate and heavy grazing by clipping the above-ground part of plants eight cm and four cm above the soil surface, respectively. The stubble height in the simulated grazing treatment was per *Gao (2008)*. Sloping parts (e.g., the sloping leaves and stems) were not straightened in the simulated

grazing treatments. We only removed plant parts that were distributed above eight cm or four cm in their natural state. The herbivory simulation was carried out three times at 45, 60, and 75 days after the rhizomes had sprouted to simulate repeated grazing in natural environments.

### *Measurements*

Although we did not assess genotypic differences among plants in our replicate pots, we nevertheless refer to each replicate potted plant as a genet. We use the term ramet to refer to a tiller that has sprouted from a rhizome bud (i.e., an individual that is a physiologically integrated component of the genet). The clipped biomass from the simulated grazing treatment was oven-dried (60 °C for 48 h, the same below) and weighed. *L. chinensis* stopped growing 90 days after the rhizomes had sprouted and the experiment was terminated. First, we measured the morphological characters of the ramets from the CK treatment, including natural height, vertical height, stem height, leaf length, leaf width, leaf number, and leaf angle. There were about 10 ramets in each flowerpot and about five leaves on each ramet. Generally, the ramet that sprouted first was the tallest and the second leaf from the ground was the longest. Thus, we selected the ramet that sprouted first in each flowerpot and chose the second leaf on the selected ramet for measurement. Leaf angle was measured using a protractor as degrees (0—90°) from the ramet stem to the measured leaf. Next, we clipped the selected ramet biomass above eight cm in height, those between 4–8 cm in height, and those below four cm in height (Note: the sloping parts of selected ramets were not straightened when clipped, and biomass data of each of the three parts were collected in their natural state), followed by oven-drying and weighing. Second, after counting the ramets, we harvested the genet above-ground biomass at different vertical distributions (i.e., above eight cm in height, between 4–8 cm in height and below four cm in height for CK and H8; above and below four cm in height for H4), followed by oven-drying and weighing. Third, all soil with roots and rhizomes in the flowerpot was transferred into mesh bags and rinsed until only the clean roots and rhizomes remained. The stem below the soil surface (ca., two cm deep) was treated as the genet above-ground biomass below four cm. We also measured the morphological characteristics of rhizomes, including length and internode number.

## Statistical analysis

Root biomass allocation, rhizome biomass allocation, and below-ground biomass allocation were estimated using root biomass divided by total biomass, rhizome biomass divided by total biomass, and below-ground biomass divided by total biomass, respectively. The above-ground biomass vertical distribution parameters, including above-ground biomass distribution below four cm and eight cm, were estimated using below 4-cm biomass divided by above-ground biomass, and below 8-cm biomass divided by total above-ground biomass, respectively. The above-ground biomass for each layer included the clipping biomass during the simulated grazing treatment.

The genet biomass allocation parameters and the above-ground biomass vertical distribution parameters were used to evaluate spatial grazing avoidance and the induced
spatial grazing avoidance using a two-way analysis of variance (two-way ANOVA). Similarly, two-way ANOVA was used to analyze the influence of long-term overgrazing-induced legacy effects on the adaptation of *L. chinensis* to grazing with respect to its growth ability (biomass accumulation), cloning ability (ramet number) and colonizing ability (rhizome length and rhizome internode number). A significant interaction between the two factors under study indicates an effect of long-term overgrazing on the response of *L. chinensis* to simulated grazing. If the data for a trait were not normally distributed or homogeneous in variance, such data were transformed using various methods (e.g., logarithmic, square root, square, reciprocal, or square root inverse rotation conversion) to attain normality and homoscedasticity. However, if data transformation could not make the data normally distributed and variances homogeneous, we conducted a one-way analysis of variance (one-way ANOVA) or Kruskal-Wallis test. If only NG or OG had a significant response to simulated grazing, this implied that responses to simulated grazing between NG and OG were disparate. In contrast, if significant differences were observed in the simulated grazing treatments in both NG and OG, we calculated the plasticity index (*PI*) to simulated herbivory of these traits using the following formula:

$$PI = (CK - H4)/CK$$

H4 was chosen for the *PI* estimate because H4 had a greater effect on *L. chinensis* than H8. Student's *t*-test or Kruskal-Wallis test was used to compare differences in the phenotypic traits of *L. chinensis* ramets collected from the different plots (i.e., NG vs OG). The relationship between the traits was explored using the Pearson correlation method.

Structural equation modeling (SEM) was used to evaluate the importance of individual morphological traits for the above-ground spatial grazing avoidance of *L. chinensis*. The model assumed that the above-ground spatial grazing avoidance of *L. chinensis*, which was measured by the above-ground biomass vertical distribution, was attributed to individual morphological characters. The Pearson correlation analysis was conducted between all parameters included in the model. The initial model was developed according to the results of the correlation analysis and basic knowledge of plant science. Furthermore, the model was modified by deleting non-significant pathways and by increasing pathways between residual variables. $\chi^2$ statistics with the associated probability, the root mean square errors of approximation with the associated probability, and the Bentler-Bonett Index or Normed Fit Index were used to evaluate the overall fit of the model.

All the analyses were completed using IBM SPSS Statistics 19.0 and the means were compared using Tukey's HSD test ($P < 0.05$). The figures were generated in Oringin2019b.

## RESULTS

### Genet biomass allocation and above-ground biomass vertical distribution

There were significant legacy effects in the biomass allocation of *L. chinensis* genets sampled from the over-grazing plot. In CK, the rhizome and belowground biomass allocation of OG decreased significantly compared with that observed from NG ($P < 0.001$), while root

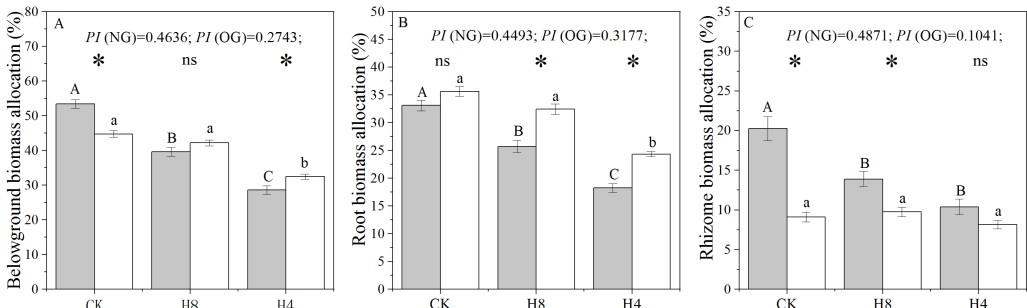

**Figure 1** **Responses of belowground biomass allocation (A), root biomass allocation (B), and rhizome biomass allocation (C) to long-term overgrazing-induced legacy effect and simulated grazing.** The grey bars indicate *L. chinensis* genets collected from the grazing exclusion plot (NG) while the white bars indicate *L. chinensis* genets collected from the continuously grazed plot (OG). CK: no simulated grazing; H8: simulated moderate grazing; H4: simulated heavy grazing. The "*" indicates a significant difference between NG and OG while "ns" indicates no significant difference. Different capital letters indicate differences between simulated herbivory treatments of NG while different lowercase letters indicate differences between simulated herbivory treatments of OG. PI is the plastic index of *L. chinensis* to simulated herbivory, estimated from (CK-H4)/CK. Two-way ANOVA was not performed due to unequal variances of these three traits.

biomass allocation was not sensitive to overgrazing-induced legacies ($P = 0.057$). Under simulated heavy grazing, OG had greater belowground biomass allocation ($P = 0.004$) and root biomass allocation ($P < 0.05$) than NG. There were significant decreases in root and rhizome biomass allocation under the simulated grazing treatment ($P < 0.05$). Compared with NG, OG showed a smaller plastic index (*PI*) under simulated grazing treatment in terms of root and rhizome biomass allocation (Fig. 1).

There was a significant difference in the genet above-ground biomass vertical distribution between NG and OG. OG tended to allocate more biomass close to the ground with a larger above-ground biomass distribution below four cm and eight cm than NG ($P < 0.001$). The simulated grazing treatment did not alter the genet vertical distribution of biomass of NG but reduced the above-ground biomass distribution of OG close to the ground ($P < 0.05$) (Fig. 2).

## Ramet above-ground biomass vertical distribution

There was a significant correlation between the ramet and genet above-ground biomass vertical distribution ($P < 0.001$) (Fig. S1). The ramet distribution below four cm and eight cm of OG were 89% and 69% larger than those of NG, respectively ($P < 0.001$). *L. chinensis* ramet morphological traits (e.g., natural height, vertical height) showed significant grazing-induced legacies under CK treatment (Table 1). Leaf angle showed the most pronounced change, increasing by 130%, while the leaf number did not respond significantly to grazing legacies (Fig. S2). Although OG ramets accumulated fewer photosynthetic products above four cm and eight cm, they had larger accumulated biomass below four cm and eight cm ($P < 0.001$) (Table 1).

SEM explained 93% and 94% of the variation in the ramet above-ground biomass vertical distribution below four cm and eight cm, respectively. The larger near-surface

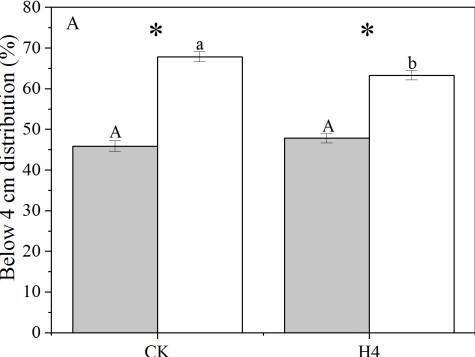
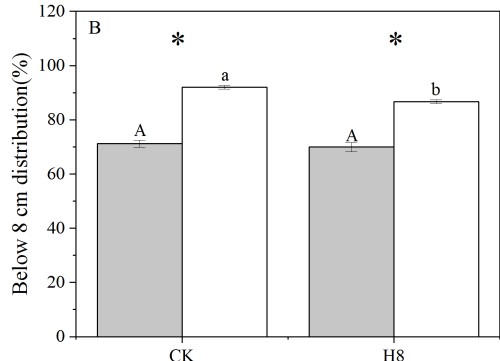

**Figure 2** **The responses of genet aboveground biomass vertical distribution to long-term overgrazing legacy effect and simulated grazing.** (A) Below four cm distribution; (B) below eight cm distribution. All parameters have the same meaning as those in Fig. 1.

**Table 1** **Long-term overgrazing-induced legacies on the ramet phenotypic traits in the control treatment (CK).** The "Leaf number" and "Leaf angle" were compared by Kruskal–Wallis test, while other phenotypic traits were compared by Student's $t$-test. Log conversion were performed for stem height, above four cm biomass and above eight cm biomass to conform to the normal distribution.

| | Means ±SE | | T(Z) | *P* |
|---|---|---|---|---|
| | **NG** | **OG** | | |
| Natural height (cm) | $21.39 \pm 0.74$ | $13.68 \pm 0.42$ | $-9.009$ | <0.001 |
| Vertical height (cm) | $22.74 \pm 0.66$ | $17.02 \pm 0.52$ | $-6.829$ | <0.001 |
| Log(Stem height (cm)) | $0.80 \pm 0.04$ | $0.61 \pm 0.02$ | $-4.507$ | <0.001 |
| Leaf length (cm) | $16.02 \pm 0.27$ | $12.92 \pm 0.37$ | $-6.767$ | <0.001 |
| Leaf width (cm) | $5.45 \pm 0.17$ | $4.97 \pm 0.17$ | $-2.047$ | 0.047 |
| Leaf number | $6.05 \pm 0.22$ | $6.41 \pm 0.20$ | $(-1.224)$ | 0.221 |
| Leaf angle | $26.60 \pm 0.69$ | $61.14 \pm 1.09$ | $(-5.793)$ | <0.001 |
| Log(Above four cm biomass (g)) | $-0.64 \pm 0.03$ | $-0.84 \pm 0.19$ | $-4.084$ | 0.001 |
| Below four cm biomass (g) | $0.05 \pm 0.00$ | $0.07 \pm 0.00$ | 4.618 | <0.001 |
| Log(Above eight cm biomass (g)) | $-0.82 \pm 0.04$ | $-1.37 \pm 0.0.08$ | $-5.878$ | <0.001 |
| Below eight cm biomass (g) | $0.13 \pm 0.01$ | $0.17 \pm 0.01$ | 4.080 | <0.001 |
| Below four cm ramet distribution (%) | $17.90 \pm 1.06$ | $33.86 \pm 2.11$ | 6.129 | <0.001 |
| Below eight cm ramet distribution (%) | $44.77 \pm 2.25$ | $75.79 \pm 2.38$ | 8.150 | <0.001 |

distribution of above-ground biomass of OG ramets stemmed from the larger near-surface biomass accumulation and smaller biomass accumulation farther from the ground, and this above-ground spacial avoidance was induced by individual morphological characteristics (Fig. 3). According to the "Standardized Total Effects," vertical height and leaf angle played a more important role than other traits in inducing the above-ground spatial avoidance below four cm, while natural height made the highest contribution to above-ground spatial avoidance below eight cm. The standardized total effects of grazing-induced legacies on the below 4-cm and 8-cm ramet above-ground biomass vertical distributions were $-0.73$ and $-0.77$, respectively (Tables S1, S2).

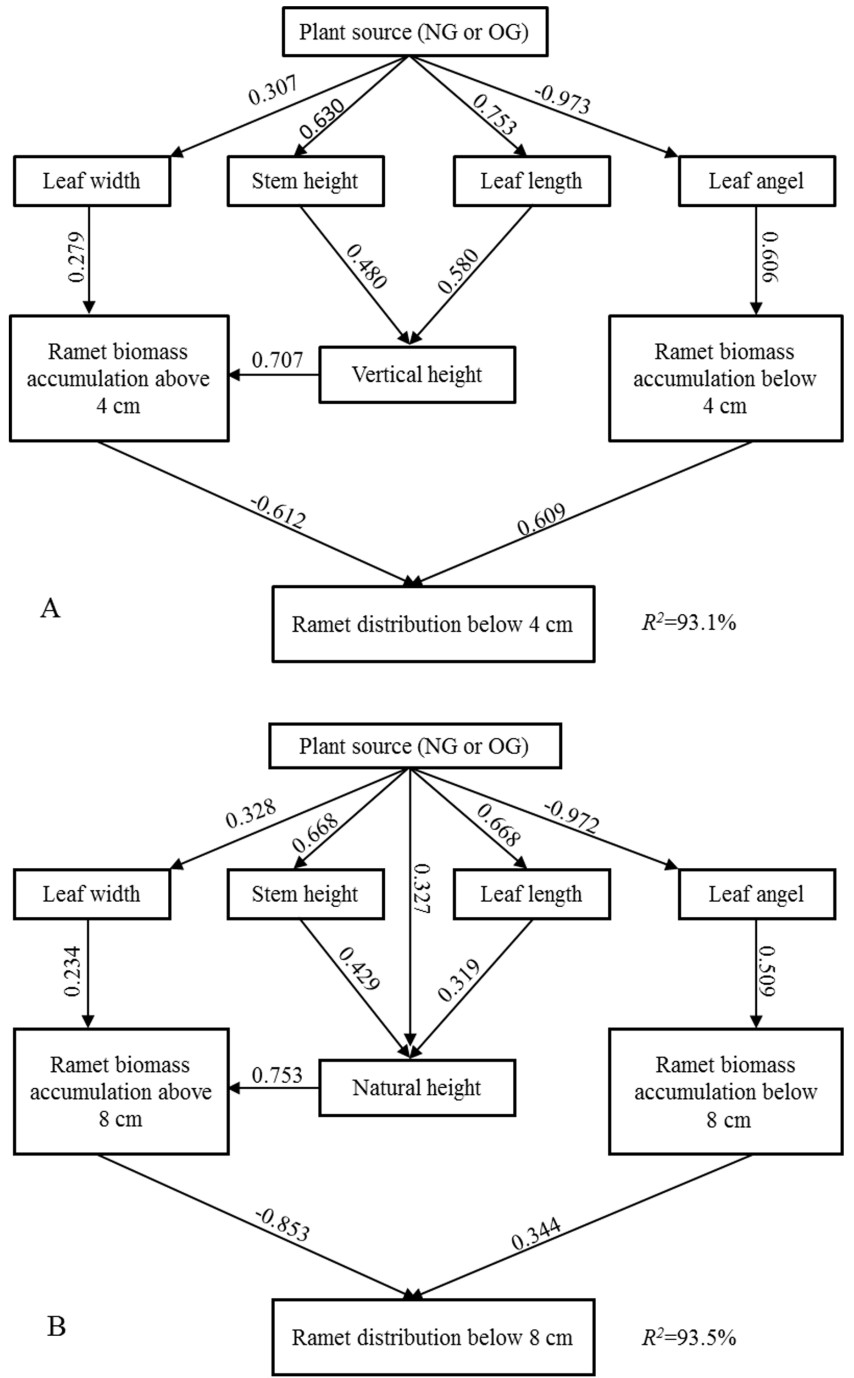

**Figure 3** **The Structural Equation Modeling (SEM) between ramet character and aboveground biomass vertical distribution.** (A) SEM in terms of below four cm ramet distribution; (B) SEM in terms of below eight cm ramet distribution.

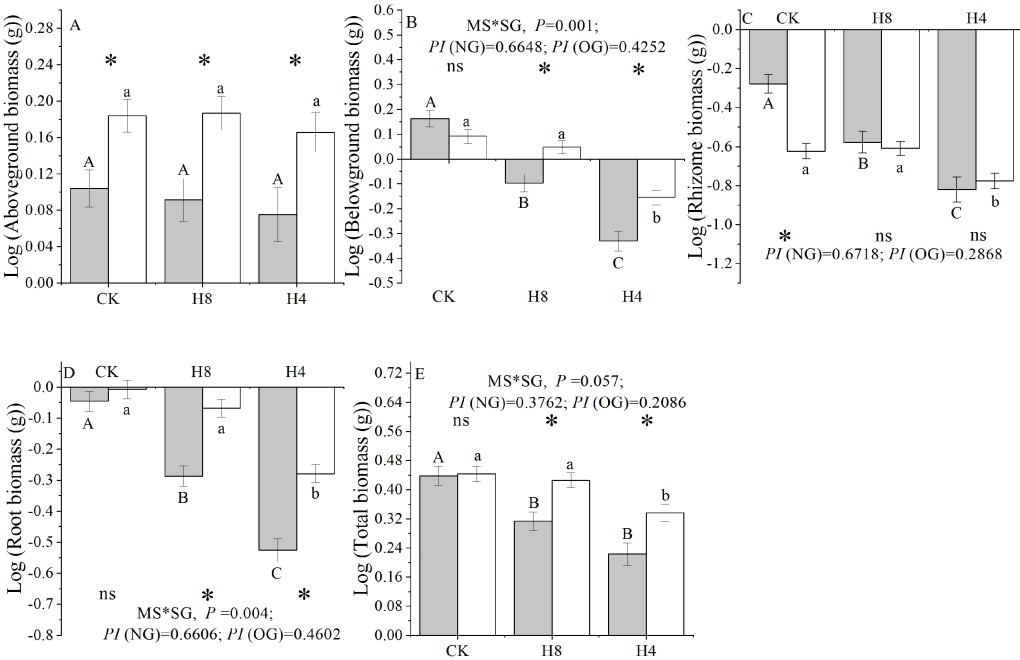

**Figure 4** **Responses of genet biomass accumulation of NG and OG to simulated herbivory.** (A) Above-ground biomass accumulation; (B) Belowground biomass accumulation; (C) Rhizome biomass accumulation; (D) Root biomass accumulation; (E) Total biomass accumulation. All parameters have the same meaning as those in Fig. 1 All these traits have been transformed logarithmically to conform to normal distribution and homoscedasticity. Two-way ANOVA was not performed due to unequal variances of rhizome biomass for above-ground biomass and rhizome biomass.

### *L. chinensis* genet performance

Simulated grazing significantly reduced the below-ground, rhizome, root, and total biomass of *L. chinensis* genets, but did not affect the above-ground biomass. NG experienced more serious biomass loss compared with OG and had a larger "*PI*" in response to the simulated herbivory in terms of the below-ground, rhizome, root, and total biomass accumulation. Additionally, there were significant interactions between plant source (NG or OG) and simulated herbivory treatment in terms of the below-ground, root, and total biomass. In the control treatment, there were no differences in the total, root, and below-ground biomass accumulation between NG and OG ($P > 0.05$). In contrast, OG accumulated higher total, root, and belowground biomass than NG under simulated herbivory ($P < 0.01$). Furthermore, NG accumulated more rhizome biomass than OG under CK treatment, while there were no significant differences observed between NG and OG under simulated grazing treatments (Fig. 4).

Simulated herbivory had no effects on ramet number ($P > 0.05$) and OG showed more ramets than NG. Simulated herbivory significantly reduced the rhizome length and rhizome internode number of NG but did not affect that of OG. Under CK, the rhizome length and the number of rhizome sections of NG were larger than those of OG ($P < 0.001$), and no differences in these variables were detected in H4 (Fig. 5).

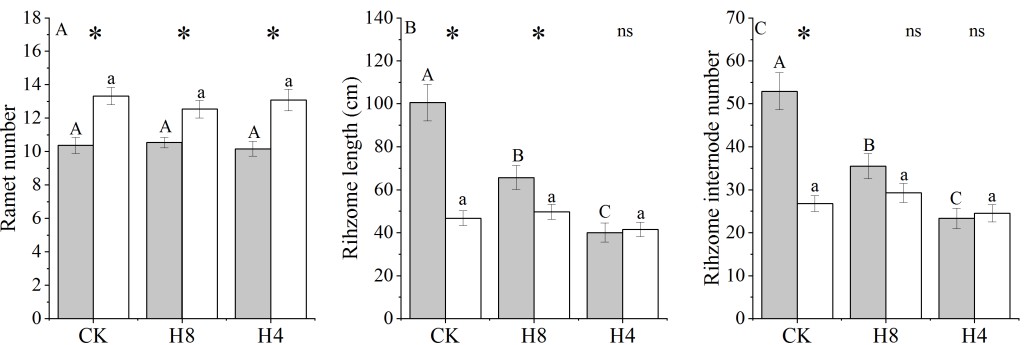

**Figure 5** **Responses of ramet number, rhizome length and rhizome internode number of NG and OG to simulated herbivory.** All parameters have the same meaning as those in Fig. 1. Two-way ANOVA was not performed due to unequal variances of ramet number, rhizome internode number and rhizome length. Because the different responses to simulated grazing of ramet number, rhizome internode number and rhizome length between NG and OG could be determined by One-Way ANOVA, *PI* of these traits were not calculated.

## DISCUSSION

### The vertical distribution of above-ground biomass

In comparison to species that decrease in abundance under intensive grazing, plant species that exhibit increase in abundance are generally shorter and more prostrate (*Diaz et al., 2007*; *Forrestel, Donoghue & Smith, 2015*). Previous work has also shown that populations of a single species may vary, with populations in over-grazed fields are characterized by more prostrate and shorter growth compared with populations in fields where grazing is excluded, and this phenomenon could persist for several generations after transplanting plants into common garden environments (*Carman, 1985*; *Polley & Detling, 1990*; *Painter, Detling & Steingraeber, 1993*; *Rotundo & Aguiar, 2008*; *Didiano et al., 2014*). Consistent with these reports, our direct measurements of the vertical distribution of above-ground biomass showed that OG allocated more above-ground biomass close to the ground. SEM analysis indicated that the shorter plant height and larger leaf angles of OG drove this pattern. This study confirms the findings of previous studies by clarifying changes in plant phenotypes under intensive grazing, specifically, more biomass is distributed close to the ground via shorter and more prostrate growth forms (*Didiano et al., 2014*).

There was no marked plasticity in the above-ground biomass vertical distribution of *L. chinensis* to simulated grazing; however, we expected that simulated grazing should significantly increase plant biomass allocation toward the ground. This might be expected given that the ramet number did not respond significantly to simulated grazing in our experiment. A large number of new shorter ramets could contribute to the near-surface allocation of plant biomass via the shorter natural height as mentioned above. On the other hand, we speculate that leaf angle, an important morphological trait influencing the above-ground biomass vertical distribution, may not exhibit significant responses to simulated grazing without grazer saliva and trampling.

Furthermore, the lack of significant responses of the vertical distribution of *L. chinensis* above-ground biomass to simulated grazing indicates that overgrazing-induced legacies in terms of the vertical distribution of above-ground biomass cannot be induced by short-term defoliation, and may be attributed instead to long-term defoliation or other pathways. In addition to defoliation, livestock can also influence plants by trampling, saliva, and indirect effects (e.g., increasing soil density, changing the rate of light interception by plants, etc.) (*Heggenes, Odland & Bjerketvedt, 2018*). Trampling may play an important role in overgrazing-induced variation in the vertical distribution of above-ground biomass. Generally, short and prostrate plants have a higher resistance to trampling than those with taller and erect growth forms (*Warwick, 1980*; *Sun & Liddle, 1993*; *Kobayashi, Ikeda & Hori, 1999*). Therefore, the procumbent growth form exhibited by OG may be related to livestock trampling. Aside from this consideration, overgrazing may promote increases in localized drought, which may be another contributor to the observed overgrazing-induced legacy effect on *L. chinensis*. Drought-adaptive morphological characteristics, such as small stature, have been reported to be advantageous for avoiding and recovering from herbivory (*Adler et al., 2004*; *Patty et al., 2010*). However, the pathways by which more above-ground biomass is allocated close to the ground as a result of overgrazing require additional research.

## Biomass allocation above- and below-ground

Contrary to our expectation, OG allocated less biomass belowground under CK than NG; this was attributed to the lower investment in rhizomes by OG. The lower biomass pre-allocation to the rhizome of OG does not benefit the herbivory tolerance and avoidance of *L. chinensis* (*Fornoni, 2011*; *Lurie, Barton & Daehler, 2017*) and can be considered a cost of herbivory tolerance in the absence of grazing (*Lennartsson, Ramula & Tuomi, 2018*). However, the reduced rhizome allocation of OG may also be the result of the trade-off between colonization and grazing tolerance, as we found a significant negative correlation between rhizome length and ramet number (Fig. S3B). Long-dispersing ramets receive less support from the mother plant (*Zobel, Moora & Herben, 2010*); hence, shorter rhizomes which are induced by the lower investment of *L. chinensis* in this organ could prevent the ramets from overgrazing-induced death. On the other hand, the shorter rhizome implies more ramets, and this could reduce the risk of ramets extinction under overgrazing.

A colonization-competition trade-off (*Herben et al., 1997*; *Zobel, Moora & Herben, 2010*; *Gough et al., 2012*) may provide another explanation for the larger investment in rhizomes observed in NG. Extreme droughts occur every few years which result in many open patches in both overgrazing and no-grazing plots (*Wang, Liu & Guo, 2019*). In the absence of grazing, plants with the most rapid colonizing ability in the open patches become dominant components of the vegetation (*Fahrig et al., 1994*; *Benot et al., 2013b*). Therefore, intermittent extreme droughts may have contributed to the longer observed rhizomes of NG. Another possible mechanism driving this phenomenon is the fact that NG plants reserved resources for ensuring the next generation of new ramets through a dense canopy and litter layer, which seriously hinders the growth and development of new ramets (*Craine et al., 2001*; *Benot et al., 2013a*).

The immediate responses of above- and below-ground allocation to simulated herbivory showed that *L. chinensis* increases the allocation of above-ground biomass, and this result supports the functional equilibrium theory (*Poorter et al., 2012*; *Gong et al., 2015*). According to this theory, more resources stored in the roots and rhizomes are mobilized to stimulate the growth of newly emerged leaves and new ramets (*Donaghy & Fulkerson, 1998*). Numerous studies have shown that this process leads to a reduced allocation of resources belowground for plants that experience leaf damage (*Gong et al., 2015*; *Barton, 2016*; *Liu et al., 2018*). However, biomass allocation of OG was less affected by simulated herbivory; this was attributed to the enhanced above-ground spatial avoidance displayed by OG, which reduced the degree of herbivory damage (Fig. S4).

## Ramet number

The stimulation of plant density through increased grazing disturbance has been reported by numerous previous studies (*Wang et al., 2017*). Concurrent to this observation, we found that there were more ramets for OG than NG, and this is consistent with previous studies (*Detling & Painter, 1983*; *Rotundo & Aguiar, 2008*). Thus, while many scientists have focused on the disruption of plant apical dominance caused by livestock defoliation (*Rautio et al., 2005*; *Lennartsson, Ramula & Tuomi, 2018*), overgrazing-induced legacy effects on ramet number could partially explain the relatively larger plant density in grazing ecosystems.

Based on the negative relationship observed between the ramet number and ramet vertical height (Fig. S3A), we speculate that the observed significant increase in the number of ramets of OG may stem from decreased apical dominance. Weak apical dominance is a coping strategy for avoiding browsing damage caused by livestock. Shorter height in plants results in superior grazing avoidance, which reduces the probability of being defoliated because more biomass is allocated close to the ground. Furthermore, a higher number of ramets could enhance tolerance to grazing by reducing the risk of ramets extinction under overgrazing. Considering this phenomenon, both grazing tolerance and grazing avoidance could thus persist simultaneously, although some studies have suggested that there is a trade-off between these two strategies (*Fineblum & Rausher, 1995*; *Mauricio, 2000*; *Krimmel & Pearse, 2016*). On the other hand, the decreased apical dominance induced by overgrazing-induced legacy effects may be attributed to an improvement in light penetration into the environment under grazing, and weaker apical dominance is most likely in the absence of competition for light (*Lennartsson, Ramula & Tuomi, 2018*).

In contrast to our prediction, we did not observe significant differences in *L. chinensis* ramet numbers between the simulated herbivory treatments in our experiment. Although many studies have indicated that plants can produce more branches when the apically dominant shoot is subjected to physical damage or herbivorous defoliation (*Klimešová & Klimeš, 2003*; *Rautio et al., 2005*; *Wang et al., 2017*), many studies have also suggested that herbivory or cutting cannot increase the ramet number (*Wang et al., 2004*; *Benot et al., 2009*) and might even reduce it (*Hicks & Turkington, 2000*). Two preconditions are required for broken apical dominance to induce an increase in ramet numbers: a sufficient number of resources and meristems for regrowth and sufficient apical suppression of basal

meristems (*Klimesova et al., 2014*; *Lennartsson, Ramula & Tuomi, 2018*). In this study, sufficient apical suppression of the basal meristems was achieved during the simulated grazing experiment. Specifically, most of the leaves were removed and only a small section of the stem was left in the "H4" treatment. Hence, there may have been limited resources and meristems for the regrowth of *L. chinensis* because of its short growth times from tiller emergence to the first simulated grazing treatment and from the last treatment to harvest.

## CONCLUSIONS

Consistent with studies on contemporary evolution and stress memory (*Detling & Painter, 1983*; *Oesterheld & McNaughton, 1988*; *Agrawal et al., 2012*; *Züst et al., 2012*; *Didiano et al., 2014*), our study showed that OG exhibited higher adaptation to simulated grazing in terms of the growth, cloning and colonizing ability than NG. This stronger adaptation was attributed to enhanced above-ground spatial avoidance. Contrary to our prediction, OG pre-allocated less biomass to the rhizome, which does not seem to promote herbivory tolerance and avoidance in *L. chinensis*; however, this also may reflect a tolerance strategy via shorter rhizomes and more ramets. Here, we quantitatively studied the grazing-induced spatial avoidance of plants for the first time and found that enhanced above-ground spatial avoidance was induced by a larger leaf angle and shorter height. However, because of the pseudo-replication and because the sample areas only consisted of two adjacent pastures in this study, the findings from our research may be limited to our study site.

## ACKNOWLEDGEMENTS

We gratefully thank Jingjing Yin and Junjie Duan for their help during the experiment.

### Funding

This study was financially supported by the Major projects of Inner Mongolia Natural Science Foundation (2020ZD06), the National Natural Science Foundation of China (31702161), the National Basic Research Program of China (2014CB138806), the Inner Mongolia Natural Science Foundation (2017MS0379), and the Inner Mongolia Natural Science Foundation (2020MS03070). There was no additional external funding received for this study. The funders had no role in study design, data collection and analysis, decision to publish, or preparation of the manuscript.

### Grant Disclosures

The following grant information was disclosed by the authors:
Inner Mongolia Natural Science Foundation: 2020ZD06, 2017MS0379.
National Natural Science Foundation of China: 31702161.
National Basic Research Program of China: 2014CB138806.

### Competing Interests

The authors declare there are no competing interests.

## Author Contributions

- Fenghui Guo conceived and designed the experiments, performed the experiments, analyzed the data, prepared figures and/or tables, authored or reviewed drafts of the paper, and approved the final draft.
- Xiliang Li and Saheed Olaide Jimoh conceived and designed the experiments, performed the experiments, authored or reviewed drafts of the paper, and approved the final draft.
- Yong Ding conceived and designed the experiments, analyzed the data, authored or reviewed drafts of the paper, and approved the final draft.
- Yong Zhang, Shangli Shi and Xiangyang Hou analyzed the data, authored or reviewed drafts of the paper, and approved the final draft.

## Data Availability

The raw measurements are available in the Supplemental Files.

## Supplemental Information

Supplemental information for this article can be found online at http://dx.doi.org/10.7717/peerj.10116#supplemental-information.

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
