# Peer review of "Overgrazing-induced legacy effects may permit Leymus chinensis to cope with herbivory"

_PeerJ, doi:10.7717/peerj.10116_

## Round 0.1 · original submission · Major Revisions

Based on the reviewers' comments, this manuscript contains some interesting findings, however major changes are required. Please carefully check the reviewers' comments. If you chose to submit a revision, it will be subject to further review before I can make a recommendation on publication.

Major issues to be addressed regarding experimental design:

1. Pseudo replication – your inferences are limited to your 2 specific study plots. This should be acknowledged.
2. Why did you select these clipping treatments (4 cm and 8 cm). How do they relate to the ecology of your study system and what is the hypothesis regarding the two different clipping heights?
3. Does clipping with scissors yield plant responses similar to sheep grazing? It may be helpful to cite other recent studies supporting this. Commonly, jasmonic acid is applied to mechanical wounds to initiate a plant response that is more similar to natural herbivory.

Methods clarifications:
Reviewer 1 points out various clarifications that are needed in the Methods. Additionally, your definition of “ramet” should be provided. Furthermore, even after viewing Fig S2, I have no idea what is meant by the terms “tiller ramet” and “rhizome ramet” (L 151). Both of them seem to be composed of tillers in Fig S2.
Why were plant parts below the soil surface treated as aboveground biomass? Cite a reference to support this.
What is the reasoning for separating biomasss layers into strata 0-4 cm, 4-8 cm, and 8+ cm? How are they predicted to differ? I understand that 0-4 cm could refer to “close to the ground” (escape from grazers?) but how are the other two strata expected to differ? Aren’t those strata both in easy reach of grazers? Assuming the sheep browse typically to ~4 cm (?), then it seems clearer to me to compare plant properties/biomass proportions < 4cm versus > 4 cm.

Manuscript Structure – All sections need attention as pointed out by Reviewer 1. Some material may be better shifted to supplemental material or cut entirely. Careful English editing is required. Please avoid the term "macro mammals"; I think it can be replaced with "grazing mammals"

Further comments:
Discussion –The Discussion has not raised the ecological concept of (over)compensation at all, although it has been suggested for other grasses subjected to grazing. Did you find any evidence in terms of biomass?
Conclusion L 351 -- What is meant by “cloning ability”? Do you mean ramet production or perhaps tiller production? Biomass production is not the same as cloning ability, and you have not presented ramet counts, so this conclusion may need to be re-thought.

Figure 2B – The y-axis should be back-transformed to match 2A.
Figure 3 – since ramet and genet distributions are correlated as shown here, it could make sense to only present one of the results in the main manuscript and move the other to supplemental materials.
Figure 6 – What is the ecological relevance of “below 8 cm”? This needs to be clarified in the text otherwise it should not be part of SEM modeling and I would also question why any data or analyses are presented on 8 cm (versus 4 cm). This study could be easier to understand if one ecologically meaningful height were chosen for analyses, corresponding to height that is often protected from sheep, if such a height exists.
Table 1 What is “Nature height”? This same term appears on L 351. Did any of the culms flower? It would be valuable to compare the height of flowering culms between grazing histories.

Reviewer 1 ·

Basic reporting

Review of Guo et al. “Overgrazing-induced legacy effects on phenotypes prepare Leymus chinensis to cope with herbivory” for PeerJ MS# 49307 June 2020
In this manuscript, Guo et al. seek to demonstrate if evolutionary adaptations/herbivory-induced legacy effects alter the phenotypes of populations of a highly palatable rhizomatous clonal grass species, Leymus chinesis, that have experienced significantly different grazing histories. Collecting samples from two pastures located in Inner Mongolian steppe grasslands that have experienced either high intensity, long-term livestock grazing or been protected from mammalian grazing they grew potted transplants in a greenhouse and subjected them to different simulated herbivory/clipping “grazing” intensities. The plants were then assessed after 90 days for differential demographic population responses based on the plants origin (legacy) and herbivory intensity treatments.
Utilizing plant material from only two different small pastures and comparing them could be considered problematic, both ecologically and statistically (pseudoreplication). Concerns regarding the latter are somewhat mitigated since the authors are primarily interested in how plant populations from these different “legacies” differentially respond to the different “grazing” intensity treatments that they experimentally implement in controlled and randomized conditions, but the former concern regarding the broader generality and applicability of these sampling areas remains.
There are a number of interesting demographic/population results reported herein, but unfortunately, the manuscript suffers from numerous shortcomings including significant weaknesses in the study design, awkward structural organization of the text, confusing conceptual development of the central themes, and issues with the interpretation and presentation of the data. Additionally, I encourage the authors to seek language editing assistance as the current grammatical state of their manuscript exceeds the reasonable editing expectations of external referees. I have made a few suggestions below throughout the manuscript, but the overall revisions needed are extensive and will require a committed editorial service. Finally, simulated herbivory studies in controlled greenhouse conditions are not novel in the ecological literature. The authors claim they are contributing information that fills a critical knowledge gap, but I, at least as presented here (and they, to an extent), am not convinced (see below). As such, I do not believe most of these concerns could be easily addressed without the revision constituting an entirely new submission.
My major concerns include:
Introduction - The introduction is poorly organized. It fails to both identify and justify the central thesis the authors seek to examine and adequately support it with previously published literature. The large paragraph describing resistance, avoidance, and tolerance is overly general and can be shortened considerably. The Methods sections suffers from numerous inadequacies and insufficient descriptions of critical procedures. Section 2.1.2. entitled “The materials sampling and cultivating” is particularly confusing and unclear.
Methods - There is little to provide confidence that the arbitrary heights of 4cm and 8cm aboveground selected to differentiate morphological adaptations to grazing have a sound ecological justification. Are there specific sheep grazing behaviors that affect suitable forage heights? Would 3cm and 7cm or 5cm and 9cm yield different results? And given that the authors merely measured the heights based on their growth in pots in artificial greenhouse conditions how would these be affected by exposure to the real-world conditions of climatic elements and animal trampling? Both here and in the Introduction, the authors fail to develop a convincing case why these results will potentially translate into ecologically meaningful patterns. Moreover, since these leaf angle and height data end up being some of the most significant results reported and strongly influence the SEM interpretation I remain underwhelmed and skeptical. There are some interesting population demography data reported, particularly the root biomass responses, but these fail to rescue the overall deficiencies noted above.
Results – The results are very dense and could be improved by adding more description with less dependence on referencing figures and tables. Indeed, based on my downstream suggestion to eliminate and/or reduce some of the numerous figures and tables, converting some of the key findings into more comprehensive or detailed text may improve the readability of this section.
Discussion – The most parsimonious explanation for L. chinensis in the OG plot pre-allocating less biomass to rhizomes, is not so much a tolerance strategy where the plant benefits from shorter rhizomes and more tiller ramets, but a cost or tradeoff associated with biomass losses and altered source-sink relationships within the plant. The authors describe this type of biomass reallocation in their Introduction and somewhat revisit it in the Discussion, but the text takes on a meandering explanation that merely leaves the reader confused. Indeed, I believe these are some of the strongest and most interesting results in the study and they merit a thoughtful and thorough interpretation in the Discussion.
Lines 254-279 – This entire section of the Discussion needs to be restructured, better clarified, and edited with improved grammar. I reread it multiple times and am still not entirely certain what takeaway message the authors are attempting to convey.
Section beginning at line 280 – there is some interesting speculation here that would be of interest to a broader base of ecological readers. Unfortunately, the authors really need field data, which they do not have, on the frequency and abundance of asexual vs sexual reproduction of the different genotypes in the different pastures to answer these questions in a satisfactory manner.
It is discouraging to read the authors state “more detailed research is needed to understand the responses of plant ramet numbers to herbivore defoliation, especially in combination with grazing-induced legacy effects on plants” at the end of their manuscript when the primary intention of their study was to make exactly this contribution! I appreciate that no study is wholly comprehensive and there are always moving goalposts on opportunities to gain more knowledge and insights, but regret that (and as the authors confess above) the current iteration of this submission falls considerably short of adding sufficiently to existing published ecological literature.
Figures & Tables - While I appreciate that supplemental electronic figures and data eliminate physical journal space restrictions and provide authors an opportunity to be more comprehensive, the cartoon drawings for Supplemental Figures 1-4 are overly simplistic and unnecessary. In their place, I recommend that several of the main figures and tables be moved to the supplemental category. Eight multi-panel figures and three data intensive tables are excessive for a study as simple and straightforward as presented here. I recommend considerably reducing the primary visual data presented to only the essential results that best tell the story the authors are crafting from their study.
A non-comprehensive sampling of other grammatical issues:
Line 12 - conditions
Line 21 - “against our prospection” is unclear. Do the authors mean “Contrary to our prediction…”?
Lines 26-27 - Awkwardly phrased sentence. Requires grammatical clarity. I also submit that the definition is either incomplete or insufficient
Line 31 - extra space after bracket
Line 32 – above the authors define legacy effects as effects on ecosystem structure and/or function, but here they refer to plant defense strategies. Plant defense strategies are conventionally thought of as plant population processes, not ecosystem process. This may seem semantic, but it is a rather critical point and influences not only the accurate usage of the legacy effect terminology, but the entire context and interpretation of it both here and as applied to other species in other ecosystems.
Lines 37-39 - You emphasize the majority of published herbivory induced legacy effects research over recent decades has focused on short-term insect damage but then fail to provide any references to support this assertion.
Line 84 – precariously?
The term “past decades” is overused in the Introduction.
Line 140 – replace “weighted with a weighing balance” with “weighed”
Line 154 – mesh bag
Line 255 – increasing trend of what?
Figure 5 legend – light

Experimental design

see above

Validity of the findings

see above

Additional comments

see above

Reviewer 2 ·

Basic reporting

no comment

Experimental design

no comment

Validity of the findings

no comment

Additional comments

The manuscript reported a novel experiment aiming to declare how the performance of Leymus chinensis in response to simulated grazing is influenced by the overgrazing-induced legacy, and what phenotypic traits contribute to the changes of the performance. The novelty of the study lies in that it not only declared the legacy effect of overgrazing by livestock rather than by insects, but declared its consequences for future coping strategies in responses to grazing. In my view, the experiment was nicely designed, the introduction section has given sufficient context of the study, the results presented are necessary and clearly stated, and the discussion section is cohesive and in depth. However, I still have some concerns as follows.

Major concerns
1. L217, when you calculated the aboveground, either for every vertical layer or the total, did you pool together all the three harvest?
2. L214-215, Why simulated grazing reduce, but not increase the aboveground biomass distribution close to the ground? As expected, the aboveground biomass distribution should be more towards ground surface in response to grazing. I cannot see the explanation for this in the Discussion section.
3. L243-245, I think this result is very interesting, and worth more deeply exploited in the Discussion section. For example, the OG plants is more conservative compared to NG plants, which are more exploitative, as shown by its greater rhizome biomass than OG plants when cultivated in CK.

Minor concerns
4. L179, “not both responded differently…” is grammatically wrong.
5. L183, there is nothing in the parentheses in the formula.
6. L207, “NG had smaller….”, can this sentence be written otherwise like: “OG had greater…..”, because intuitively, the readers will take NG as a control to OG, so you’d better use OG as the subject of the sentence, and compare with NG. The same problem with the sentence in L209-210.
7. L218, here the authors mentioned Fig.3, but in my view, this figure is of little significance, because genet is composed of ramets, genet and ramet should have the same biomass distribution.
8. L269, “……may not be ascribed to defoliation by livestock.” I don’t agree. Over-grazing induced legacy is formed during a long-term period through many generations. It is a bit arbitrary to conclude that “the over-grazing induced legacies distribution may not be ascribed to defoliation by livestock” only based on one-off and short-term test.
9. L298-301, this explanation is a bit far-fetching. In addition, I am afraid that the vegetative tillers sprouted from the rhizome nodes cannot termed as seedlings, as they were not originated from seeds.
10. Since you have define the codes like OG and NG, you should use them when necessary, to avoid redundancy. See Line 308-310 in the annotated manuscript. Please make the similar changes where else when necessary throughout the entire manuscript.
11. I would term the OG plot overgrazed plot instead of overgrazing plot, and term the NG plot ungrazed plot instead of no grazing plot.
12. Figure 2, I would like the two panels combine into one panel (of course you should first do the same data transformation for below 4 cm distribution as for below 8 cm distribution).
13. Figure 7, GZ in the parentheses should be OG?

Please see attached annotated manuscript for other very minor corrections

Annotated reviews are not available for download in order to protect the identity of reviewers who chose to remain anonymous.

---

## Round 0.2 · Major Revisions

This manuscript has been substantially improved. However,I have identified additional additional necessary as follows:
Title – I don’t think the current wording is supported by the available data. Replacing “permit” with “may permit” would be improve accuracy.
L 22 “cloning ability” – the term needs to be clearly defined or avoided (see further comments below)
L 25 “while height” to “while reduced tiller natural height”
L 28 “via shorter rhizome and more tiller ramets.” to “where reduced allocation to rhizomes is associated with increased production of ramets”.
L 43 “the plant-herbivore relationship” to “plant-herbivore relationships”
L 43 “adaptability” to “adaptations”
L 44 “the interspecific relationship, and” to “as well as interspecific relationships and”
L 77 “the common garden” to “common garden”
L 97 “grazing disturbance” to “disturbance”
L 123 “Given that the field site from which we collected materials was not free from pseudo-replication,” to “Although we acknowledge pseudoreplication in the experimental design, as each treatment consisted of one large plot with subsamples as replicates,”
L 127 “and to decrease the impacts of pseudo-replication on the experimental results.” to “across each of the large neighboring plots.”
L 130 what does “well-prepared” mean?
L 131 “Materials cultivation in the greenhouse” to “Cultivation in the greenhouse”
L 132 “into 2-cm long” to “ into a 2-cm long section”
L 152 “the soil surface.” to “the soil surface, respectively”.
L 154 “in its” to “in their”
L 156 “imitate” to “simulate”
L 158 I think more explanation / justification is needed for these terms. To start, I suggest something like “Although we did not assess genotypic differences among plants in our replicate pots, we nevertheless refer to each replicate potted plant as a genet. We use the term ramet to refer to a tiller that has sprouted from a rhizome bud (i.e. an individual that is a physiologically integrated component of the genet).” I find the following sentence (L 159) very confusing . The problem is that a tiller is simply defined as a grass stem. Therefore, what you have drawn on Fig S1 as “rhizome ramet” is actually a tiller (individual grass stem). I think new terms may be needed here, and just as importantly, please explain to readers why it is important or interesting to separate these two types of tillers. I suppose one type contributes to spatial spread while the other favors increasing biomass density. Is this difference relevant to your study on grazing effects? After reading the entire manuscript, I see that you have not described differences between ramet types in Results or discussed the two ramet types in Discussion (except brief mention of “tiller ramets" on L 365), so I wonder if confusion could be avoided by combining all ramets rather than defining two types? See also comments on Fig 5 below.
L 173 oven-dring to oven-dryng
L 177 “the mesh bag” to “mesh bags”
L 193 “respected to” to “respect to”
L 194 “rhizome section number” – what does it mean?
L 194 Cloning ability if used here, needs to be clearly defined. Rhizome length doesn’t seem to relate to cloning ability, rather it relates to physical spread.
L 201 What is GZ?
L 204 “plasticity index to simulated herbivory of these traits (PI)” to “plasticity index (PI) to simulated herbivory of these traits”
L 207 Why does this formula have “( )” to the right of each variable?
L 207 Why was H4 selected for the PI?
L 218 What does “increasing pathways” mean?
L 226 Due to the pseudoreplicated design, this line should be worded something like “There were significant legacy effects in the biomass allocation of L. chinensis genets sampled from the over-grazing plot.”
L 228 “and root biomass” to “while root biomass”
L 241-242 round off percent values here to whole numbers (89%, 69%, 130%)
L 244 What does “homogenous environment” mean here?
L 245 It is impossible for any value to decrease by more than 100%. If a value decreases by 100% it means there is nothing left.
L 247 “it had” to “they had”
L 262 “genet” to “genets”
L 263 What is meant by “disturbances”? Biomass loss?
L 263 “based on” to “in response to”
L 264 “Besides” to “Additionally”
L 277 “under H4” do your mean “for H4” or “in H4”?
L 280 “Compared with” to “In comparison to”
L 281 “an increasing trend in” to “increase in”
L 282 “one species population” to “populations of a single species may vary, with populations”
L 285 “the common garden” to “common garden”
L 289 “smaller” to “shorter”
L 299 “may also do not” to “may not”
L 300 “saliva” to “grazer saliva”
L 309 “high” to “taller”
L 322 “overgrazing-resulted more severe drought” to “overgrazing may promote increases in localized drought, which” [I think this is what you mean?]
L 325 I don’t think “viability” is the correct word. Do you mean “aboveground growth and grazing tolerance” or something similar?
L 331 “ Colonization-competition” “ A colonization-competition”
L 333 “resulted in” to “which result in”
L 362 “be stem” to “stem”
L 374 “In Contrast” to “In contrast”
L 394 “benefit the” to “seem to promote”
L 394 “of L. chinensis” to “in L. chinensis”
L 395 “shorter rhizome” to “shorter rhizomes”
L 399 delete “small”
Fig 1 caption – delete “light-dark”
“the L. chinensis genet collected from grazing” to “L. chinensis genets collected from the grazing”
“the L. chinensis genet collected from” to “L. chinensis genets collected from”
Fig 3 – For “material source” I suggest “plant source (NG or OG)” to increase clarity.
Fig 5 It is not clear how these 5 graphs relate to “cloning ability”. Perhaps production each new ramet could be considered a new clone that is integrated as one plant; however, how does rhizome length indicate cloning ability? The “total ramet production” graph could be presented in place of the so called “tiller ramet” and “rhizome ramet” graphs as the same statistical findings and trends are the same.
Table 1 caption “while oter phenotypic plastics” what does this mean? Do you mean PI? I don’t see PI in the table.
Define “Lg” in the table.

---

## Round 0.3 · Minor Revisions

The revised text is generally clear and assertions are supported by the data. I identified a few minor edits as follows:
L 83 delete “highly” (relished already implies this so “highly” is redundant)
L 129 “of the buds” to “containing buds”
L 165 “ramet” to “ramets”
L 198 “homogeneous” to “homogeneous in variance”
L 208 can you add a sentence “H4 was chosen for the PI estimate because H4 had a greater effect on L. chinensis than H8.”
L 281 “Besides, previous work has shown” to “Previous work has also shown”
L 282 “over-grazing fields are more prostrate and shorter” to “over-grazed fields are characterized by more prostrate and shorter growth”
L 283 “with that in fields” to “with populations in fields”
L 319 “pre-allocating biomass” to “biomass pre-allocation”

---

## Round 0.4 · accepted · Accept

Minor edits as follows:
Line 115 "studying sites" to "study sites"
L 120 "used the high-" to "used high-"
L 271 "differences were observed" to "differences observed"
L 281 "DÍAZ et al" [fix capitalization]
L 283 "fields are characterized" to "fields being characterized"